# Peroxidase-Mimicking Ir-Te Nanorods for Photoconversion-Combined Multimodal Cancer Therapy

**DOI:** 10.3390/nano13111706

**Published:** 2023-05-23

**Authors:** Gyeonghye Yim, Seounghun Kang, Subean Kim, Hongje Jang

**Affiliations:** 1Department of Chemistry, Kwangwoon University, 20 Gwangwoon-ro, Nowon-gu, Seoul 01897, Republic of Korea; 2Department of Chemistry, Seoul National University, Seoul 08826, Republic of Korea

**Keywords:** iridium–tellurium nanorods, galvanic replacement, nanozymes, photoconversion, cancer therapy

## Abstract

Owing to multiple physicochemical properties, the combination of hybrid elemental compositions of nanoparticles can be widely utilized for a variety of applications. To combine pristine tellurium nanorods, which act as a sacrificing template, with another element, iridium–tellurium nanorods (IrTeNRs) were synthesized via the galvanic replacement technique. Owing to the coexistence of iridium and tellurium, IrTeNRs exhibited unique properties, such as peroxidase-like activity and photoconversion. Additionally, the IrTeNRs demonstrated exceptional colloidal stability in complete media. Based on these properties, the IrTeNRs were applied to in vitro and in vivo cancer therapy, allowing for the possibility of multiple therapeutic methodologies. The enzymatic therapy was enabled by the peroxidase-like activity that generated reactive oxygen species, and the photoconversion under 473, 660 and 808 nm laser irradiation induced cancer cell apoptosis via photothermal and photodynamic therapy.

## 1. Introduction

Nanoparticles composed of a variety of elements are promising candidates for diverse applications. The advanced form of nanoparticle is a core–shell nanostructure with distinct internal and external components. Depending on the composition, the indirect properties imparted by the inner constituent elements and the direct surface properties imparted by the outer shell constituent parts can exhibit a synergistic effect. As previously described for core–shell nanoparticles, the strata-specific morphology with a few nanometer-scale shells maintains the properties of a metallic core composition while simultaneously incorporating the properties of a shell composition [1,2,3,4,5]. Galvanic replacement is a promising technique for incorporating additional elements into the metal nanoparticle template. Moreover, by successfully mixing heterogeneous metal compositions, the plasmonic effect of the metal nanotemplates can be maintained [6,7,8,9].

As nanoparticles facilitate multifunctional therapeutics on a physicochemical level, cancer therapy is one of the most promising fields for the application of nanoparticles with a variety of elemental compositions [7,10,11,12]. The conjugation of anticancer agents on the surface of nanoparticles is frequently performed in various chemical processes [13,14,15,16]. This strategy is effective for clearly targeted cancer therapy that does not adversely affect other major organs [17,18] because free anticancer drugs frequently induce a variety of adverse effects in vivo [19,20]. Among the physical approaches, photothermal (PT) therapy (PTT) utilizing the plasmonic property of a metallic composition possesses significant potential. PTT is accomplished via near-infrared laser irradiation, which causes the localized surface plasmon resonance to generate heat sufficient to induce apoptosis in cancer cells [21]. Owing to the relatively large bandgap of certain nanoparticles ranging from 1–4 eV, laser irradiation with a shorter visible wavelength enables photocatalytic conversion [22]. This property leads to the conversion of nearby oxygen and water molecules to reactive oxygen species (ROS), hydrogen peroxide, hydroxyl radical, and superoxide radical [23,24,25]. However, using a laser for targeted cancer therapy has limitations due to the limited penetration depth of the wavelength of each laser. For instance, photodynamic (PD) treatment with a visible-wavelength laser is ineffective outside the laser’s penetration range of a few millimeters below the skin [26]. However, combining phototherapy with biochemical strategies may be highly effective in overcoming the limitations of cancer phototherapy.

Recently, nanozymes, comprising nanoparticles and enzymes, have been demonstrated to be a viable therapeutic technique. ROS is generated when enzymes, such as peroxidase, catalase, oxidase, and superoxide dismutase, repeatedly transfer electrons between water and oxygen molecules. Moreover, ROS induces apoptosis [27,28,29]. Compared to some nanoparticles, nanozymes successfully replicate natural enzymes and are even more advantageous in terms of the viability of enzymatic characteristics across a wide range of temperature and pH than natural enzymes composed of proteins [30,31]. Nanozymes, which can alter the microenvironment in tissues and cells or trigger responses to reactive chemical species, have recently gained interest as an effective strategy in combinatorial cancer treatment with phototherapy [32,33,34,35]. In the current study, we propose the use of iridium–tellurium nanorods (IrTeNRs) in cancer therapy (Figure 1). The IrTeNRs were synthesized via hydrothermal galvanic replacement using Te nanorods (TeNRs) as a sacrificial template. IrTeNRs possessed a hollow interior structure and a rough surface structure. It exhibited high absorption across the ultraviolet (UV)-visible (Vis)-near-infrared (NIR) spectral regions, indicating efficient PT conversion induced by laser irradiation at various wavelengths. Additionally, the surface IrO_x_ composition exhibited photocatalytic activity independent of PT conversion, indicating that photocatalytic and PD efficiencies varied with wavelength. Additionally, the photo-nanozymatic cancer therapeutic effect of the IrTeNRs was established in vitro and in vivo by its low toxicity to cells, blood, and organs, in addition to the efficient photo-triggered reaction and exceptional peroxidase (POD)-like activity. Compared with the more commonly used elemental nanoparticles, the use of IrTeNRs is a novel strategy, and it is anticipated that Ir-based medicinal applications possess considerable promise for future therapeutics.

## 2. Results and Discussion

IrTeNRs were successfully synthesized in this study via galvanic replacement (Figure 1). Owing to the difference in the standard reduction potentials of Te (Te^4+^|Te^0^ = 0.57 V vs. standard hydrogen electrode (SHE)) and Ir (Ir^3+^|Ir^0^ = 1.156 V vs. SHE), the as-prepared TeNRs were partially oxidized and sacrificed for the reductive growth of Ir (on surface sedimentation). The conversion of TeNRs to IrTeNRs was confirmed using the UV-Vis spectral analysis. The extinction peak of TeNRs at 674 nm was previously derived from the peak fluctuation caused by the transition from the p lone-pair valence band of Te to its p* conduction band [36]. After galvanic replacement, the newly added metallic Ir aided the hypsochromic shift of the peak to the ultraviolet region (Figure 2a). The UV absorption band between 350 and 400 nm can be attributed to water ligands coupled with Ir to enable spin-allowed π → π* transitions [37].

IrTeNRs were characterized in detail using transmission electron microscopy (TEM) images. Slightly curved nanorods with lengths of approximately 300–400 nm were observed owing to the insertion of the reduced Ir0, disrupting the hexagonal lattice structure of the metallic Te. Hollow nanorods, which are the most distinctive feature of galvanic replacement nanostructures, were expected to form because the TeNRs used as templates dissolved owing to the oxidation of the inner section (Figure 2b). The high-resolution TEM (HR-TEM) images clearly demonstrated their hollow structure, which was decorated with small spherical Ir nanoparticles via hydrothermal regrowth (Figure 2c). Additionally, using energy dispersive spectrometry, high-angle annular dark field scanning TEM (HAADF-STEM) revealed the elemental compositions by detecting signals from Ir Lα (red) and Te Lα (yellow) (Figure 2d). Ir and Te were miscible throughout the IrTeNRs, according to EDS mapping results.

X-ray analysis was used to further characterize the composition and crystallinity of the sample. X-ray photoelectron spectroscopy (XPS) was used to interpret the oxidation states of each constituent element. Considering the binding energy of the deconvoluted Te 3d peak, Te existed as metallic Te^0^ (583.1 and 572.8 eV for 3d_3/2_ and 3d_5/2_, respectively), oxidized TeO_2_ (585.3 and 574.9 eV), TeO_3_ (586.9 and 576.6 eV), and suboxides (573.8 eV for 3d_5/2_) (Figure 3a) [38]. Owing to the absence of an additional reducing agent during synthesis, the reduced telluride form (Te^2−^) was not evolved, and a significant amount of oxidized Te was presumed to have formed during storage and sample preparation following synthesis. In the case of Ir, the major state was metallic Ir^0^ (63.8 and 60.8 eV for 4f_5/2_ and 4f_7/2_, respectively), and the sample surface was simultaneously deposited with oxidized Ir^4+^ (64.7 and 61.8 eV) (Figure 2b) [39]. The wide scan XPS survey spectrum revealed that IrTeNRs are composed entirely of Ir and Te with a variety of orbital binding energy peaks (Appendix A).

X-ray diffraction (XRD) was used to verify the crystal structure of IrTeNRs. Various peaks corresponding to numerous atomic planes were observed for rutile IrO_2_, including those at 39.9° (200), 54.6° (211), and 56.2° (220). Additionally, at 40.8° (111) and 47.4° (111), the peaks related to metallic Ir were observed (200) [40]. The embedded Te^0^ crystallinity was determined using the peaks at 31.0° (102); 40.0° (110); 46.3° (003); 47.6° (200); and 56.2° (202) (Figure 3c) [41,42,43]. Sharp and intense signals at 32.8° (211) and 44.3° (220) were attributed to Si (211) and Si (220), which originated from the IrTeNR film-supporting materials [44].

Because IrTeNRs exhibited broad UV-Vis-NIR absorption, PT conversion efficiency was evaluated at wavelengths of 808 nm (T_f, 808_ = 45.8 °C, ΔT_808_ = 22.9 °C), 660 nm (T_f, 660_ = 37.0 °C, ΔT_660_ = 14.1 °C), and 473 nm (T_f, 473_ = 32.5 °C, ΔT_473_ = 9.6 °C) using diode laser irradiation-mediated PT conversion and temperature elevation in a quartz cuvette system (Figure 4a). Contrary to the negligible heat dissipation observed with the 1× phosphate-buffered saline (PBS) control (T_f, 1xPBS_ = 23.7, ΔT_1xPBS_ = 0.8 °C), significant heat dissipation was observed during the total 600 J irradiation of the laser output. The trend confirmed that even with the same energy, a longer wavelength resulted in a more effective PT conversion. When temperature elevations were observed for various concentrations of IrTeNRs exposed to 808 nm irradiation, a proportional relationship was observed (Figure 4b). Additionally, there was no fluctuation during the 808 nm laser on–off cycling test when the temperature was repeatedly increased and decreased, demonstrating the thermal stability of IrTeNRs (Figure 4c). Furthermore, the PT conversion efficacy of the IrTeNRs was measured to be 64.7 % (Figure 4d).

Photocatalytic activity against photon energy transfer was expected to develop owing to the semiconducting IrO_2_ composition. To demonstrate the photocatalytic potential of IrTeNRs, we performed the photocatalytic reactions, type-I (generation of reactive oxygen radical species) and type-II (generation of singlet oxygen), under irradiation with various wavelengths. The photodegradation of methylene blue (MB) identified the PD type-I pathway. The highest decomposition of MB was observed at 473 nm (33.9%), after 600 J of irradiation, followed by 660 nm (30.1%) and 808 nm (19.0%) (Figure 4e). Type-II exhibited a similar trend of demonstrating a higher PD potential at a shorter wavelength. The fluorescence signal from the singlet oxygen sensor green, singlet oxygen indicator, was the strongest at 473 nm (F/F_0_ = 3.82), followed by 660 nm (3.43) and 808 nm (2.19), indicating that the photocatalytic efficacy is higher at shorter wavelengths than that at longer wavelengths (Figure 4f). In both cases, there was no change in the 1xPBS control. The contrasting efficiency dominance between PT and PD therapies is interpreted as the difference in efficiency between the direct use of a wavelength corresponding to an energy near the bandgap energy and plasmon-like hot electron transfer.

The colorimetric detection of 3,3′,5,5′-tetramethylbenzidine (TMB) against a variety of factors, including the concentration of H_2_O_2_, IrTeNRs, and TMB as well as pH, confirmed the POD-like activity of IrTeNRs (Figure 4g and Appendix A). In all cases, at pH 5, the absorbance at 652 nm wavelength, where the oxidized TMB (TMB_ox_) absorbs, increased as the concentration of each compound increased. Among the compounds, the absorbance increase was the most sensitive to the TMB concentration gradient. The different colors from each pH condition were successfully detected because the color is dependent on the conjugating tendency of TMB_ox_. Moreover, the single diazotized TMB_ox_ appeared yellow at an acidic pH, while the TMB_ox_ appeared blue at pH 5 (optimized) [30].

Prior to evaluating the photo-nanozymatic cancer therapeutic potential in vitro, the colloidal stability in various physiologically buffered solutions was determined in vitro. IrTeNRs exhibited excellent colloidal stability in DI water and 1xPBS at all tested time points (0 to 24 h), as determined using UV-Vis spectra and digital photo images (Appendix A). IrTeNRs were well dispersed without coagulation or sedimentation in Dulbecco’s Modified Eagle Medium (DMEM) and protein-containing complete cell culture media (Appendix A). The biocompatibility of IrTeNRs was determined using a cell viability assay against human cervical cancer cells, specifically HeLa cells (Appendix A). IrTeNRs exhibited no severe toxicity at concentrations less than 2.5 μg Ir/mL nor dose-gradient cytotoxicity at concentrations higher than 2.5 μg Ir/mL. Therefore, the safe dose of IrTeNRs was fixed at 1.5 μg Ir/mL for evaluating the efficacy of PTT and PD therapy (PDT) in cell-based experiments.

The same characteristics as those ascertained in the cuvette systems were observed in the live/dead staining used to confirm cancer cell apoptosis due to PDT and PTT. At 4 °C, PDT alone induced significant cancer cell apoptosis (473 nm > 660 nm > 808 nm). In comparison, at a long wavelength (808 nm > 660 nm > 473 nm), PTT-only treatment with ROS scavenger incorporation (L-histidine) induced significantly more cancer cell apoptosis. In the laser-irradiated region, the PDT and PTT combination therapy induced apparent cell death at all wavelengths (Appendix A).

To quantify the IrTeNR-mediated nanozymatic enhancement of POD-like activity in cell-based tests, HeLa cells were pre-treated with or without IrTeNRs and then incubated with various concentrations of H_2_O_2_. Cell viability was determined using the MTT assay (Figure 4h). When cells were co-treated with IrTeNRs and H_2_O_2_, a clear gradient of cytotoxicity was observed in response to the H_2_O_2_ concentration, owing to the POD-like activity of IrTeNRs in the cell cytoplasm. By contrast, when cells were treated with H_2_O_2_ alone, no cytotoxicity was observed. The intracellular ROS level was determined using 5-(and-6)-carboxy-2′,7′-difluorodihydro-fluorescein diacetate (Carboxy-H_2_DFFDA) indicators; the ROS level significantly increased in cells co-treated with IrTeNRs and H_2_O_2_. When IrTeNRs or H_2_O_2_ were used in isolation, the ROS level did not significantly increase in comparison to 1xPBS (control group) (Figure 4i). This demonstrates that IrTeNRs retain POD-like activity in the cytoplasm and induce apoptosis in cancer cells via intracellular H_2_O_2_.

The in vitro quantification of the combinatorial therapeutic efficacy was performed using cell viability assay followed by fluorescent microscopic observation with live/dead staining and Hoechst 33,342 nuclei staining. (Figure 5a). PDT demonstrated an increasing trend of cell viability—24.0%, 37.5%, and 62.2% under 473 nm, 660 nm, and 808 nm irradiated conditions, respectively, whereas PTT demonstrated a decreasing trend in the order of 62.3%, 58.9%, and 15.8%, respectively, for the wavelengths of 473 nm, 660 nm, and 808 nm. PTT and PDT, when used in combination, significantly enhanced therapeutic effect by further decreasing cell viability to 9.4%, 9.4%, and 11.8% under 473 nm, 660 nm, and 808 nm irradiated conditions, respectively (Figure 5b).

Prior to conducting the preclinical sensory evaluation, hemolysis and inductively coupled plasma mass spectrometry (ICP-MS) analyses of organs (heart, liver, lung, spleen, kidney, and tumor) were performed to confirm the acute blood toxicity and in vivo distribution of IrTeNRs. Even at the maximum dose (80 μg Ir/mL), IrTeNRs did not cause severe hemolysis (Appendix A). To verify in vivo distribution, 200 μL of IrTeNRs (100 μg Ir/mL) was intravenously injected into the tail vein of mice, and 24 h later, the major organs (heart, liver, lung, spleen, and kidney) and tumor tissue were harvested for analysis. The liver (17.89 μg) accumulated the most IrTeNRs injected into the mice, followed by the tumor (3.55 μg), kidney (2.23 μg), and spleen (2.15 μg) (Appendix A).

The in vivo sensory evaluation of the IrTeNR-based therapeutic approach was conducted in accordance with the preclinical trial protocol. For tumor implantation, HeLa cells were subcutaneously injected into the right flank of five-week-old BALB/c nude male mice. When the tumor volume reached 50 mm^3^, the tail vein was injected with 1xPBS (control) and IrTeNRs (2.0 μg Ir/g mice weight) at a 6 d interval (day 1, 5, 11, and 17). After 24 h, the tumor was irradiated with an 808 nm laser (2 W/cm^2^) (Figure 6a). Prior to conducting a full-scale evaluation, it was established that IrTeNR-mediated phototherapy could induce temperature elevation and cancer cell apoptosis when exposed to an 808 nm laser. According to infrared thermographic images, when mice injected with IrTeNRs were irradiated with the laser, the temperature of the tumor site was significantly increased above 54 °C, which could result in irreversible heat shock to the cancer cells. In comparison, no significant increase in temperature was observed in mice injected with 1xPBS under the same laser-irradiated condition (Figure 6b). This result demonstrated that IrTeNRs accumulated in the tumor via passive targeting systemic delivery were sufficient for PTT when exposed to laser light, and IrTeNR-based phototherapy could inhibit tumor growth.

When tumor growth was observed, NPs alone (28.2-fold) inhibited 35.5% of tumor growth via POD-like activity, compared to 1xPBS (43.6-fold) and NIR alone (39.6-fold) control groups. PDT alone (22.1-fold) and PTT alone (16.9-fold) demonstrated a tumor suppression effect of 49.3% and 61.2%, respectively, compared to NPs alone. The final strategy, a combination therapy (10.1-fold), demonstrated an excellent tumor growth inhibition effect of 76.8% (Figure 6c). Histopathology examinations, including hematoxylin and eosin (H&E) staining and terminal deoxynucleotidyl transferase dUTP nick and labeling (TUNEL) staining for further investigation of the therapeutic efficacy, revealed that the tumors treated with NPs alone demonstrated mild tissue disruption and apoptosis in cancer cells. When compared to NPs alone, the PDT- and PTT-only treatments caused significant tissue disruption and apoptosis in cancer cells, whereas the combination therapy caused significant tissue disintegration and apoptosis in cancer cells throughout the tumor region (Figure 6d).

To assess the potential risks associated with nanocarrier-mediated cancer therapy, a toxicological profile of therapeutic groups (NPs alone, PTT, PDT, and PTT and PDT) and control groups (1xPBS and +NIR only) was obtained. Serological analysis revealed similar levels of expression of the nanoparticle-induced hepatic toxicity indicators aspartate aminotransferase (AST) and alanine aminotransferase (ALT), renal injury and toxicity indicators blood urea nitrogen (BUN) and creatinine (Crea), and chronic toxicity and cell membrane damage markers total protein (TP) and lactate dehydrogenase (LDH) (Appendix A). In hematology, there were no differences in white blood cell, red blood cell, or platelet counts between the therapeutic approach groups and control groups (Appendix A). As with the previous blood-analysis-based toxicity evaluation, no significant pathological abnormalities were observed in the therapeutic approach groups when compared to the control groups in histopathology (Appendix A). Additionally, no discernible change in the sensible mice body weight was observed in the experimental groups during the preclinical trial (Appendix A). When combined with toxicological and therapeutic data, IrTeNRs appear to be a promising candidate with excellent biocompatibility suitable for in vivo application and photo-thermal/dynamic combination therapy utilizing IrTeNRs effectively suppresses tumors.

## 3. Conclusions

Using TeNRs as a sacrificing template, this study successfully synthesized IrTeNRs via galvanic replacement. IrTeNRs exhibited POD-like activity in the presence of H_2_O_2_, corroborating the possibility that IrTeNR-induced cancer cell apoptosis occurs spontaneously in the presence of H_2_O_2_. When combined with PTT and PDT, IrTeNRs demonstrated excellent in vitro tumor treatment efficiency, with cancer cells remaining at 9.4% after laser irradiation at 473 nm and 660 nm, respectively. The same trend of multimodal cancer treatment being significantly more effective than single-modal cancer treatment was observed in vivo from the tumor growth deterioration with 76.8% inhibition efficiency and tumor tissue death using the TUNEL assay. During therapy, no physiological damage to significant organs or abnormal body weight loss were detected.

## 4. Experimental

### 4.1. Materials

Sodium tellurite, iridium(III) chloride, hydrazine hydrate (52–56%), polyvinylpyrrolidone (PVP, MW = 40 kDa), 3-(4,5-dimethylthizol-2-yl)-2,5-diphenyltetrazolium bromide (MTT), doxorubicin, protease inhibitor cocktails, phosphatase inhibitor cocktails, and TUNEL assay kits were purchased from Sigma (St. Louis, MO, USA). Ethylene glycol and buffer solutions with a pH range of 1–7 were purchased from Samchun (Pyeongtaek, Gyeonggi-do, Republic of Korea). 3,3′-5,5′-Tetramethylbenzidine (TMB) was purchased from Alfa Aesar (Ward Hill, MA, USA). Hydrogen peroxide (30%), dimethyl sulfoxide (DMSO, 99%) and sodium hydroxide were purchased from Duksan (Ansan, Gyeonggi-do, Republic of Korea). Dulbecco’s modified eagle’s medium (DMEM), 10× phosphate-buffered saline (PBS), fetal bovine serum (FBS), penicillin-streptomycin, and 0.05% Trypsin-EDTA were purchased from WelGene Inc. (Deagu, Republic of Korea). Live/dead viability/cytotoxicity assay kit and singlet oxygen sensor green were purchased from Molecular Probes Invitrogen (Carlsbad, CA, USA).

### 4.2. Synthesis of IrTeNRs

A previously described procedure was used to prepare the sacrificial TeNR nanotemplate [27]. In brief, 92.2 mg of sodium tellurite was dissolved in the 40 mL of ethylene glycol with 1 g of PVP (Mw 40 kDa), and 0.5 mg of NaOH in the transparent glass vial. To the homogeneous mixture, 1.3 mL of hydrazine monohydrate was injected and heated for 3 h under vigorous stirring at 70 °C. Subsequently, 20 mL of the as-prepared TeNR solution was homogeneously mixed with 2.5 mL of 10 mM IrCl_3_ in a 50 mL Teflon-lined autoclave vessel. After natural cooling, the stainless-steel reactor containing the Teflon-lined autoclave was heated at 200 °C for 2 h without stirring, and a dark-brown solution was obtained as the final product. The solution was then purified using centrifugation with DI water for at least 3 times, with 10 min per washing at 9000 rpm. The precipitate was finally re-dispersed in 20 mL of DI water, and the concentration of the dispersion has been denoted as 1 equivalent (eq).

### 4.3. Characterization

The size and shape of IrTeNRs were determined using an energy-filtered TEM LIBRA 120 (Carl Zeiss, Germany), a Cs-corrected STEM JEM-ARM200F (JEOL, Japan), and a high-resolution field emission SEM SU8010 (JEOL, Japan) (Hitachi, Japan). The X-ray photoelectron spectrometer system K-Alpha+ (ThermoFisher Scientific, USA) and SmartLab were used to conduct the analysis (Rigaku, Japan). Lambda-465 (PerkinElmer, Waltham, MA, USA) and SynergyMx were used to obtain UV-Vis spectra (Biotek, UK). An SOLC laser (Shanghai Laser and Optics Century Co., Ltd., Shanghai, China) was used for 473 and 660 nm irradiations, while OCLA surgical laser accessories were used to obtain the 808 nm laser irradiation (Soodogroup Co., Pusan, Republic of Korea). Images of in vivo thermography were captured using an FLIR one pro (FLIR System, Wilsonville, OR, USA).

### 4.4. Peroxidase-like Activity

UV-Vis spectroscopy was used to conduct a colorimetric analysis of POD-like activity. To prepare the reaction solutions, 1 eq. IrTeNR solution was diluted to 0.25 eq. and TMB was diluted in DMSO to a concentration of 10 mM. Thereafter, 10 μL of each solution was added to a 2 mL buffer solution, which was placed in a disposable plastic cuvette. After adding 10 μL of 30% H_2_O_2_ to the solution, UV-Vis spectra was recorded every 10 s for 3 min to detect changes in the color of the solution. To determine the sensitivity of [H_2_O_2_], [particle], and [TMB], the concentrations of these components were varied by adding 5 μL, 10 μL, and 20 μL of the respective solutions during each step. To investigate the color diversity as a function of pH, we used a variety of buffer solutions with pH values ranging from 1 to 7.

### 4.5. Photothermal Conversion Efficiency Measurement

Temperature elevation was used to confirm photothermal conversion efficiency when various concentrations of IrTeNRs were irradiated using a laser at wavelengths of 473, 660, and 808 nm. Each nanorod-dispersed aqueous solution was placed in a quartz cuvette and irradiated with a total of 600 J of laser energy. Temperature changes were recorded using a digital thermometer at each of the 120 J irradiation points to allow for easy comparison.

### 4.6. Photocatalytic Activity Measurement

For type-I ROS sensing, 1 μL of 5 mM SOSG in methanol was added to 1 mL of IrTeNRs (2.5 mg Ir/L) in 1xPBS. The solution was irradiated with 473, 660, and 808 nm laser at 4 °C with sealing to prevent the evaporation by involved PT. The SOSG fluorescence intensity of the solution was measured at every 120 J irradiation point until it reached 600 J (Ex: 504 nm and Em: 525 nm).

For type-II ROS sensing, 1 μL of 10 mM MB in DI water was added to 1 mL of IrTeNRs (2.5 mg Ir/L) in 1xPBS. The solution was irradiated with the laser in the same manner as that for type-one ROS sensing. The absorbance of MB was determined at every 120 J irradiation until it reached 600 J. The absorbance at 665 nm was determined using a UV-Vis spectrophotometer.

### 4.7. Colloidal Stability of IrTeNRs

To test the colloidal stability, 2.5 mg Ir/L of IrTeNR solution in DI water, 1xPBS, serum-free DMEM media, and serum-containing DMEM media were incubated at 37 °C. At 0, 12, and 24 h, colloidal stability of each nanoparticle was determined using a UV-Vis spectrophotometer.

### 4.8. Cytotoxicity of IrTeNRs

MTT powder was dissolved in 1xPBS and filtered through a sterilized syringe filter at 5 mg/mL concentration (0.2 mm pore diameter). An MTT stock solution was prepared and stored at 4 °C. HeLa cells were seeded at a density of 10,000 cells per well in a 96-well culture plate containing 100 μL growth media (50–70% confluency). Cells were treated with appropriate concentrations of IrTeNRs in serum-containing media and incubated at 37 °C for 24 h. Following incubation, the cells were rinsed twice with 1xPBS, and 100 μL of serum-free cell media containing 0.5 mg/mL MTT was added. After an additional incubation for 2 h, the cells were washed once with 1xPBS. Subsequently, each well received 100 L DMSO to dissolve the water-insoluble formazan salts. Subsequently, a plate reader was set to 560 nm and used to determine the optical density. Thereafter, the mean and standard deviation of the triplicated data were determined and plotted.

### 4.9. In Vivo Toxicity of IrTeNRs

Following the preclinical trial, the mouse was sacrificed for blood and organ harvesting to determine the toxicity profile of IrTeNRs. During the sacrifice procedure, mouse blood was collected. Whole blood was collected in an EDTA-treated tube, and serum was collected in a serum-separating tube. Using FUJI DRI-CHEM FDC3500 and Hemavet 950, the toxicity profile was determined (AST, ALT, BUN, creatinine, total protein, lactate dehydrogenase, red blood cells, white blood cells, and platelets).

### 4.10. Hemolysis Assay of IrTeNRs

For hemolysis assay, 1 mL of the mouse whole blood was added to 14 mL 1xPBS and centrifuged for 5 min at 9500 rpm. The process of washing was repeated five times. Thereafter, 15 mL of 1xPBS was used to disperse the washed red blood cells. A 0.2 mL of red blood cell solution was added to 0.8 mL IrTeNRs in 1xPBS, and the nanoparticle–red blood cell mixture was incubated under following conditions: 90 rpm, room temperature, and in a dark place. The positive and negative controls for the hemolysis assay were DI water and 1xPBS, respectively. After 4 h, the mixture was centrifuged for 3 min at 9500 rpm. Afterward, 0.2 mL of the supernatant from the mixture was transferred to 96-well plates, and the absorbance of hemoglobin at 577 nm and a reference at 655 nm were determined.

### 4.11. Cell Based Nanozyme Activity Test

To confirm the peroxidase-like activity, 100 L of IrTeNRs (2.5 mg Ir/L) in serum-free cell media was treated with HeLa cells seeded at a confluence of 10,000 cells per well in 96-well plates. After 6 h of incubation, the particle-containing cell media was removed and washed twice with 1xPBS, followed by 3 h of incubation with 100 L of H_2_O_2_ (0.1, 0.2, 0.4, 0.8 M). The HeLa cells were then rinsed twice with 1xPBS, followed by the addition of 100 L of complete cell media and incubation for an additional 12 h. Thereafter, the cells were subjected to cell proliferation and MTT assays.

### 4.12. Cell-Based Photo Thermal/Dynamic Therapy

To demonstrate the PDT effect of IrTeNRs, 100 μL of IrTeNRs (2.5 mg Ir/L) in serum-free cell media was used to treat HeLa cells seeded at a confluence of 10,000 cells/well in 96-well plates. After 6 h of incubation, the particle-containing cell media was removed and washed twice with 1xPBS, followed by 100 μL of complete cell media replacement. The HeLa cells were then irradiated with 360 J of 473, 660, and 808 nm laser light at 4 °C for an additional 12 h to prevent cell death due to PTT. Thereafter, the cells were rinsed twice with 1xPBS,, and the cell proliferation and MTT assays were performed.

To confirm the PTT effect of IrTeNRs, 100 μL of IrTeNRs (2.5 mg Ir/L) in serum-free cell media was used to treat HeLa cells seeded at a confluence of 10,000 cells/well in 96-well plates. After 6 h of incubation, the particle-containing cell media was removed and washed twice with 1xPBS. Subsequently, 100 μL of L-histidine (20 mM) was added to the serum-containing cell media and incubated for 1 h. The HeLa cells were then irradiated at room temperature with 240 J of 473, 660, and 808 nm laser. After removing the L-histidine-containing cell media and washed twice with 1xPBS, the serum-containing cell media was replaced. The subsequent procedure was identical to that used to confirm the PD therapeutic effect.

### 4.13. In Vivo Cancer Photo Thermal/Dynamic Therapy

All animal experiments were conducted in accordance with the guidelines established by Seoul National University’s Institutional Animal Care and Use Committee (IACUC) (SNU-190612-2). Male BALB/c nude mice (5–8 weeks old) were obtained from ORIENT BIO (Sungnam-si, Republic of Korea). HeLa cells (1 × 10^6^ cells) in 100 μL 1xPBS into the right flank were subcutaneously injected to make the mouse tumor model. When the tumor reached ~100 mm^3^ in size (8~10 days after the tumor inoculation), 100 μL of 1xPBS (as a control) and IrTeNRs (80 mg Ir/L) were intravenously injected on day −1 and repeatedly injected with 6 days intervals (day 5, 11, and 17). After 24 h of the injection, the mouse tumors were irradiated for 120 s with an 808 nm NIR diode laser (OCLA surgical laser system, Soodogroup Co., Pusan, Republic of Korea) at a power density of 2 W/cm^2^. During the irradiation, the tumor temperature was measured using a thermography camera (FLIR one pro, FLIR System, Wilsonville, OR, USA). Prior to laser irradiation, 100 μL of L-histidine (20 mM) was injected into the tumor for PTT-only test. The laser was irradiated for 20 s and then rested for 40 s to prevent the temperature increase in the tumor and induction of apoptosis in cells. Every two days, the tumor size was determined using the equation 1/2 × longest diameter × (shortest diameter)^2^.

### 4.14. Histological Evaluation

Histological samples were obtained during the sacrifice procedure. For 24 h, samples from the heart, lung, liver, spleen, kidney, and tumor were immersed in a 4% PFA solution. Thereafter, the samples were embedded in paraffin and sectioned to a thickness of 10 μm. The pathology center stained the sectioned samples with H&E, IHC, and TUNEL (College of Medicine, Seoul National University, Republic of Korea). The specimens were examined using a BX71 microscope equipped with a 20× objective lens (Olympus, Tokyo, Japan).

## Figures and Tables

**Figure 1 nanomaterials-13-01706-f001:**
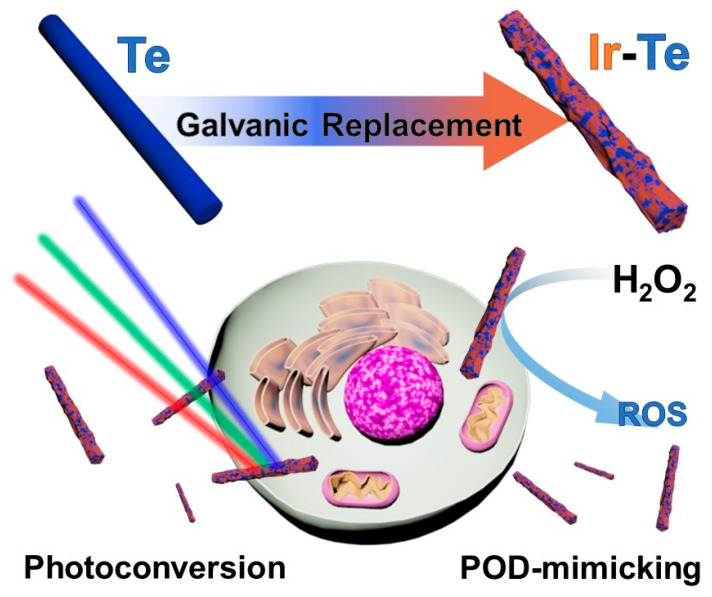
Schematic illustration of IrTeNR synthesis and photo-nanozymatic multimodal cancer therapeutic application.

**Figure 2 nanomaterials-13-01706-f002:**
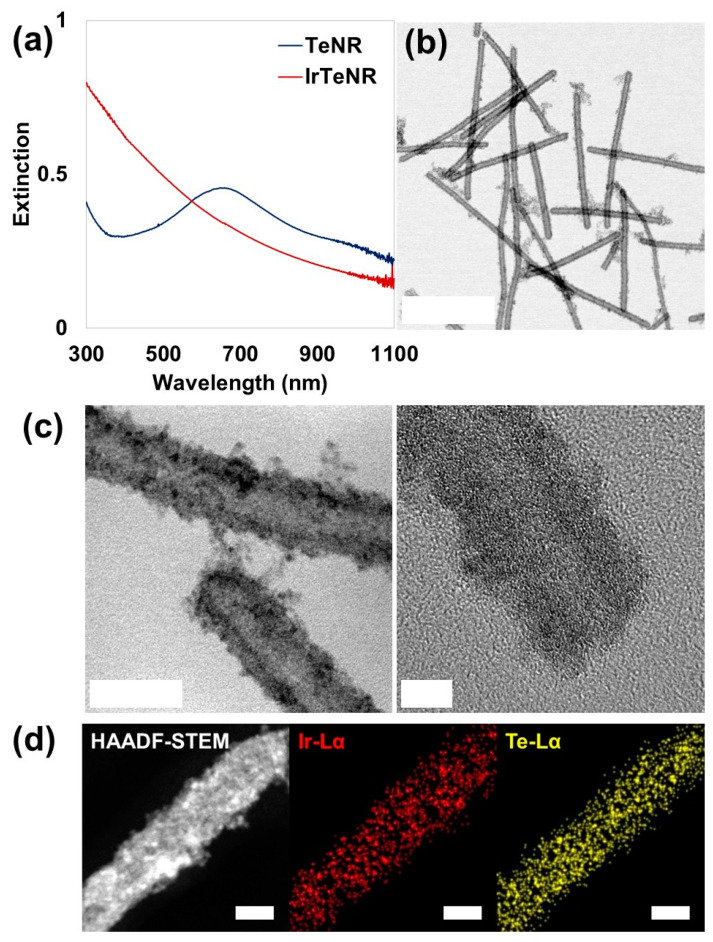
Characterization of IrTeNRs. (**a**) UV-Vis-NIR spectra of TeNRs and IrTeNRs. IrTeNRs showed excellent absorbance with a tendency to gradually decrease over the entire UV-Vis-NIR region. (**b**) Normal TEM and (**c**) HR-TEM images of IrTeNRs. Scale bars are 200, 20, and 5 nm for (**b**), (**c**) (left), and (**c**) (right), respectively. (**d**) HAADF-STEM/EDS mapping images. Scale bar is 10 nm.

**Figure 3 nanomaterials-13-01706-f003:**
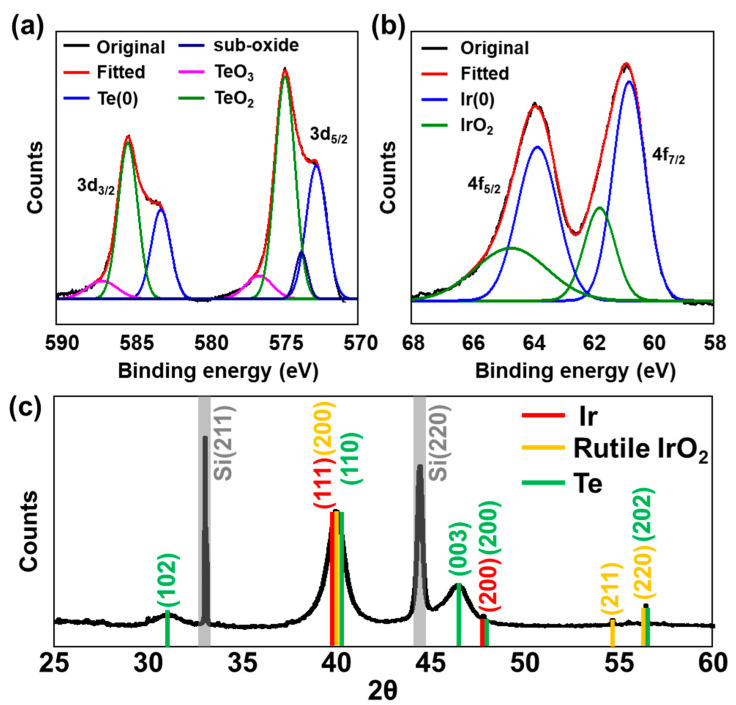
X-ray characterization of IrTeNRs. Deconvoluted XPS spectra of (**a**) Te 3d and (**b**) Ir 4f. (**c**) XRD pattern of IrTeNR films on the Si wafer.

**Figure 4 nanomaterials-13-01706-f004:**
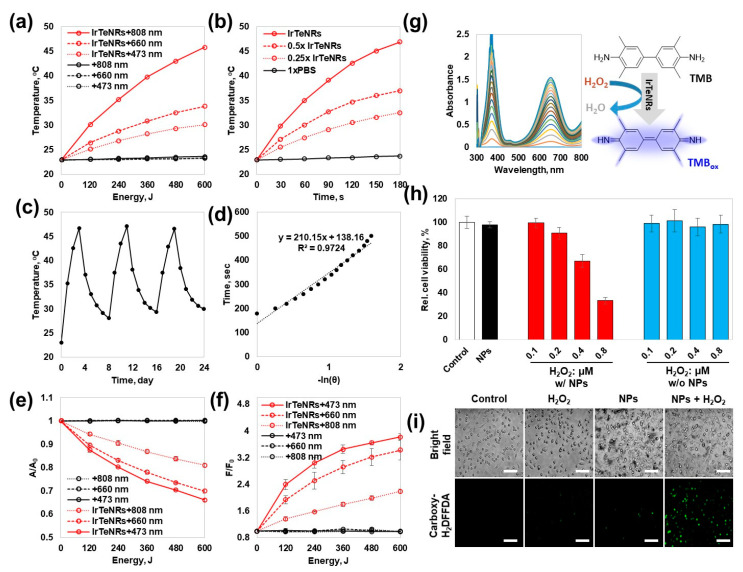
PT, PD, and POD efficiency of IrTeNRs. Temperature elevation measurement to confirm the PT conversion with different (**a**) laser wavelength and (**b**) IrTeNR concentration. (**c**) On–off cycling test using 808 nm laser and (**d**) heating–cooling-mediated PT efficiency plot for IrTeNRs. Photocatalytic activity identification for (**e**) PD type-I and (**f**) Pd type-II pathways. (**g**) POD-like activity measurement via TMB oxidation. (**h**) In vitro nanozymatic cancer therapy and (**i**) intracellular radical generation by IrTeNRs and H_2_O_2_. Scale bars are 50 μm.

**Figure 5 nanomaterials-13-01706-f005:**
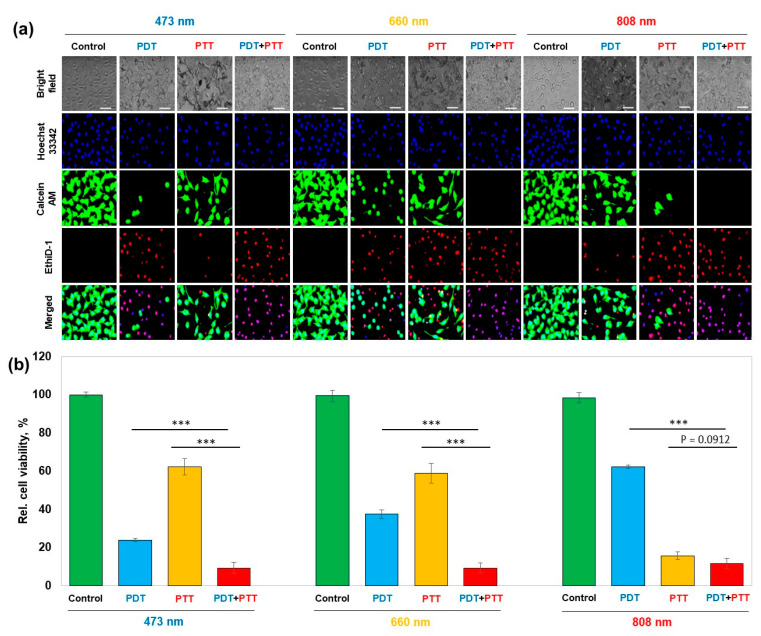
In vitro quantification of PTT and/or PDT efficacy with nanozymatic enhancement against HeLa cells. (**a**) Fluorescent microscope images for PTT and/or PDT under the irradiation of 473, 660, and 808 nm laser. Scale bars are 50 μm. (**b**) Relative cell viability under various test conditions. Statistical analysis was performed by one-way ANOVA. (*** *p* < 0.001).

**Figure 6 nanomaterials-13-01706-f006:**
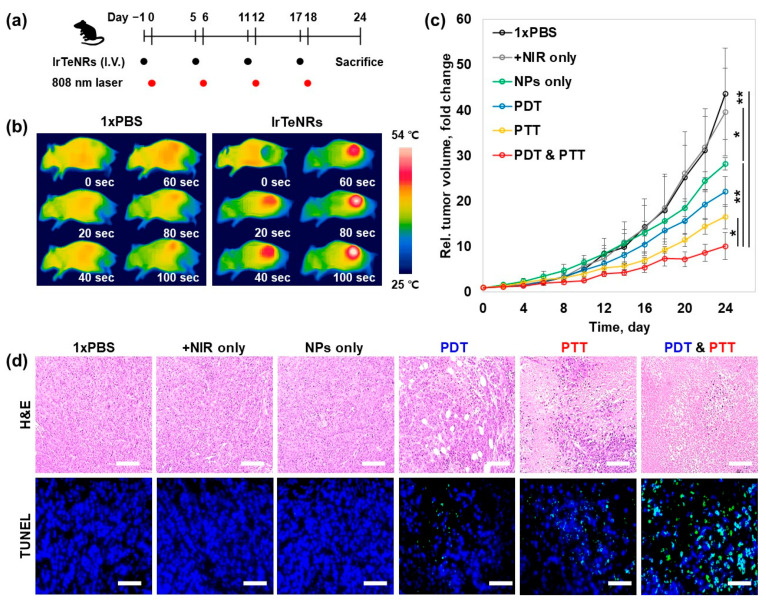
In vivo combinatorial cancer therapy with IrTeNRs. (**a**) Scheme of the pre-clinical trial schedule for tumor implantation and laser irradiation. (**b**) Comparison of in vivo heating between the injection of 1xPBS and IrTeNRs using thermography camera (FLIR one pro). (**c**) Relative tumor volume measurement (* *p* < 0.05 and ** *p* < 0.01). (**d**) Histological observations after H&E staining and TUNEL assay. Scale bars are 50 μm.

## Data Availability

The data presented in this study are available on request from the corresponding author.

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
