# Peer review of "Peroxidase-Mimicking Ir-Te Nanorods for Photoconversion-Combined Multimodal Cancer Therapy"

_nanomaterials, 2023, doi:10.3390/nano13111706_

Round 1

Reviewer 1 Report

The paper reports on the preparation of Ir-Te nanorods by galvanic replacement technique and their detailed study including the analysis of peroxidase-like activity and photoconversion property as well as the evaluation of their possible application in anti-cancer therapy (both in vitro and in vivo). The results obtained are new and worth of being published. The subject of the paper fits well the scope of Nanomaterials journal.

I have the following comments:

1. I would suggest adding a paragraph in the Introduction section to provide a brief overview of existing nanozymes and their uses in anti-cancer therapy.

2. Please provide UV-Vis-NIR absorption spectra for IrTeNRs to confirm their broad UV-Vis-NIR absorption.

3. Please provide dynamic light scattering data and zeta-potential measurements to confirm the colloidal stability of nanorods in various media.

4. In the Experimental section of the manuscript, please provide a brief description of IrTeNRs synthesis procedure.

Author Response

Thank you for sending the reviews on the manuscript referenced above. We are pleased with the comments and suggestions of editor and reviewers, and have made changes to the manuscript accordingly, as detailed are included in response letter.

Reviewer 2 Report

The manuscript is beautifully designed and presented

The idea is clear and well described by experimental data.

In general, the manuscript is written very well, in detail and with high quality.
All obtained results are confirmed experimentally.
It is necessary to make minimal technical changes in the materials and methods section.

1. Provide a detailed description of work in animals
2. Figure 5 It is necessary to make and indicate the statistical difference
3. Figure 6 b – there is no information about the device that was used to detect the animal's body temperature, and there is also no information in the materials and methods.

Author Response

Thank you for sending the reviews on the manuscript referenced above. We are pleased with the comments and suggestions of editor and reviewers, and have made changes to the manuscript accordingly, as detailed in response letter.
